# Treatment Strategy for Ultra-High-Risk Multiple Myelomas with Chromosomal Aberrations Considering Minimal Residual Disease Status and Bone Marrow Microenvironment

**DOI:** 10.3390/cancers15092418

**Published:** 2023-04-22

**Authors:** Kazuhito Suzuki, Shingo Yano

**Affiliations:** Division of Clinical Oncology and Hematology, Department of Internal Medicine, The Jikei University School of Medicine, 3-19-18 Nishi-Shimbashi, Minato-ku, Tokyo 105-0003, Japan

**Keywords:** multiple myeloma, ultra-high-risk chromosomal abnormalities, minimal residual disease, treatment

## Abstract

**Simple Summary:**

Proteasome inhibitors, immunomodulatory drugs, anti-CD38 monoclonal antibodies (triple class drugs), and autologous stem cell transplantation (ASCT) are promising myeloma treatments that have resulted in minimal residual disease (MRD) negativity and improvement in the bone marrow microenvironment. In the MASTER trial, a four-drug induction therapy, comprising triple-class drugs followed by ASCT, resulted in MRD negativity and long progression-free survival in patients with standard- and high-risk cytogenetics; however, it proved insufficient to overcome the poor outcomes in patients with ultra-high-risk chromosomal aberration (UHRCA). Hence, MRD negativity in autografts could signal positive clinical outcomes after ASCT. Additionally, high-risk myeloma cells lead to poor clinical outcomes owing to aggressive myeloma behavior as well as a poor bone marrow microenvironment. Thus, several new approaches for treating patients with MRD-positive UHRCA myeloma before and after ASCT are needed.

**Abstract:**

Despite the development of anti-myeloma therapeutics, such as proteasome inhibitors, immunomodulatory drugs, anti-CD38 monoclonal antibodies, and autologous stem cell transplantation (ASCT), multiple myeloma remains incurable. A trial treatment combining four drugs—daratumumab, carfilzomib, lenalidomide, and dexamethasone—followed by ASCT frequently results in minimal residual disease (MRD) negativity and prevents progressive disease in patients with standard- and high-risk cytogenetics; however, it is insufficient to overcome the poor outcomes in patients with ultra-high-risk chromosomal aberration (UHRCA). In fact, MRD status in autografts can predict clinical outcomes after ASCT. Therefore, the current treatment strategy might be insufficient to overcome the negative impact of UHRCA in patients with MRD positivity after the four-drug induction therapy. High-risk myeloma cells lead to poor clinical outcomes not only by aggressive myeloma behavior but also via the generation of a poor bone marrow microenvironment. Meanwhile, the immune microenvironment effectively suppresses myeloma cells with a low frequency of high-risk cytogenetic abnormalities in early-stage myeloma compared to late-stage myeloma. Therefore, early intervention might be key to improving clinical outcomes in myeloma patients. The purpose of this review is to improve clinical outcomes in patients with UHRCA by considering MRD assessment results and improvement of the microenvironment.

## 1. Introduction

Multiple myeloma (MM) is an incurable hematologic malignancy, although the development of proteasome inhibitors (PIs), immunomodulatory drugs (IMiDs), autologous stem cell transplantation (ASCT), and monoclonal antibody (MoAb) drugs has prolonged survival in patients [1]. Previously, we reviewed the literature regarding the total therapy strategy, including PIs, IMiDs, anti-CD38 MoAb, and ASCT, which not only induce a therapeutic effect on the myeloma cells but also improve the bone marrow microenvironment, including the enhancement of anti-myeloma immunological activity and the suppression of inhibitory anti-myeloma immunological effects [2,3]. Furthermore, it has recently been suggested that minimal residual disease (MRD) negativity is a surrogate marker for prolonged survival [4,5]. In fact, several studies have shown that myeloma cells detected as MRD may develop drug resistance and affect the surrounding immune environment. 

The Monoclonal Antibody-Based Sequential Therapy for Deep Remission in Multiple Myeloma (MASTER) trial, a phase II study, demonstrated that a four-drug combination of daratumumab (DARA), carfilzomib (CFZ), lenalidomide (LEN), and dexamethasone (DEX) (D-KRd) followed by ASCT and subsequent D-KRd consolidation therapy is effective for patients with newly diagnosed MM (NDMM). Once a patient entered an MRD surveillance (MRD-SURE) phase, after bone marrow samples tested MRD-negative twice consecutively, the treatment was discontinued [6]. In the MASTER study, MRD-SURE was achieved in most patients, and the survival outcome was excellent in patients with zero or one high-risk chromosomal aberration (HRCA). However, patients with two or more HRCAs, who were considered to have ultra-high-risk chromosomal aberrations (UHRCAs), not only had a low MRD negativity rate but also achieved MRD-SURE relapse in some cases. That is, the MASTER trial results suggest that other strategies are necessary for patients with UHRCAs, even if the total therapeutic strategy concerning the bone marrow microenvironment was performed and MRD negativity was achieved [6].

In this review, we discuss the treatment strategies for patients with MM and UHRCAs. Specifically, we analyze the clinical limitations of the MASTER trial in four parts: first, the genetic background of patients with UHRCAs; second, the necessity of MRD negativity in autografts; third, the method of MRD measurement; fourth, which strategies should be used moving forward to improve the prognosis of patients with UHRCAs. In particular, we discuss strategic therapies, including new targeted agents and preemptive interventions against high-risk smoldering MMs (SMMs) before they become UHRCAs.

## 2. Genetic Background of Patients with UHRCAs in the MASTER Trial

In the MASTER trial, the MRD negativity rate and survival time were lower in patients with UHRCAs than in the other chromosomal aberration (CA) groups. Of the 123 patients who participated in the MASTER trial, 24 (20%) were classified as having UHRCAs, including 83% with 1q21 gain or amplification (amp), 75% with deletion del(13q), 58% with del(17p), 54% with translocation t(4;14), and 17% with t(14;16). Herein, we discuss 1q21 gain or amp and del(17p), which were the most frequently observed chromosomal abnormalities in the MASTER trial [6]. Of the 20 patients with UHRCAs, 6 relapsed during treatment; all these patients had chromosome 1q21 aberrations and 5 had del(17p). These two CAs are occasionally detected as additional events for IgH chromosomal translocations or hyperdiploidy, which are considered an initial event for myelomagenesis and are identified less frequently in early-stage myelomas, such as monoclonal gammopathy of undetermined significance (MGUS) and SMM, than in symptomatic myeloma [7]. In addition, these two CAs are associated with refractory myeloma cells after treatment with PIs and IMiDs [8]; an increasing incidence of these two CAs has also been reported in double refractory MM, which is refractory to both PIs and IMiDs [9]. 

The 1q21 gain and del(17p) aberrations reportedly exhibit genomic instability. Recently, NEIL1, which acts at a cell cycle checkpoint and is associated with DNA repair, was reported as a novel target of myeloma cells with 1q21 gain [10]. The incidence of 1q21 gain is positively correlated with del(13q) and t(4;14) and negatively correlated with t(11;14) [7]. A key reason for this is that +1q occurs as a secondary event, often in conjunction with other high-risk cytogenetic abnormalities and genomic complexity, complicating the search for the main drivers of outcomes and response to therapy. 

There are various candidate target genes for 1q21 gain, such as *CKS1B* and *MCL1* [11]. Overexpression of *CKS1B* induces an aggressive clinical course as it promotes myeloma cell growth by activating cyclin-dependent kinases [12,13], leading to upregulation of the STAT3 and MEK/ERK pathways [14]. Meanwhile, *MCL1* is a member of the BCL2 family of anti-apoptotic proteins, with a genetic locus at 1q21 [15]. Upregulated *MCL1* expression is highly correlated with the presence of +1q among patients with NDMM [16]. *MCL1* dependency is also enhanced by IL-6 signaling within the bone marrow microenvironment [17]. 1q21 gain negatively correlates with the gene expression level of tumor necrosis factor receptor-associated factor 3 (*TRAF3*), a control protein of the nuclear factor kappa B (NF-κB) pathway, inducing refractoriness for IMiDs [18]. In the ENDURANCE trial, which directly compared the effectiveness of KRd and VRd for non-transplant-eligible NDMM, PFS was longer in the KRd group compared with the VRD group among patients with 1q21 gain and without 1q21 amp [19]. This finding indicates that CFZ improves the clinical outcome in patients with 1q21 gain compared to BOR. 1q21 gain also induces the JAK-STAT pathway, which reduces CD38 expression in myeloma cells [20]. Therefore, the efficacy of anti-CD38 MoAb may be relatively low in patients with 1q21 gain [21]. Meanwhile, anti-CD38 MoAb could contribute to myeloma cell death by not only binding to myeloma cells but also several immune cells, leading to reduced abundance of regulatory T cells (Tregs) and activation of cytotoxic T lymphocytes and natural killer (NK) cells [22]. Therefore, anti-CD38 MoAb could be considered as induction therapy for NDMM even if 1q21 gain is present. Finally, 1q21 gain could be one of the causes of extramedullary disease (EMD) during relapse [23]. Thus, 1q21 gain is a CA that has recently received particular attention owing to its associations with poor prognosis and other HRCAs, which is called double-hit MM [24,25]. In a recent single-cell study, myeloma cells with 1q21 gain worsened the immune system by increasing the prevalence of tumor-associated macrophages (TAMs), decreasing the activity of effector NK cells and the gene burden of *SLAMF7*, leading to an aggravating immune microenvironment and the proliferation of myeloma cells [26].

Clinically, in patients with NDMM and 1q21 gain or amp, the response to bortezomib (BOR), LEN, and DEX (VRd) treatment is better than in patients without 1q21 gain or amp; however, the progression-free survival (PFS) and overall survival (OS) are shorter [27]. In addition, a retrospective study has reported poor prognosis for MM in patients with 1q21 gain at a cutoff value of 6% [28], although the cutoff of numerical abnormalities is generally defined as 8–20% [29], and a poor prognosis is determined when 1q21 gain is identified not only in the main clones but also in subclones [30]. In the FORTE trial, the PFS in patients with 1q21 gain was similar to that in those without 1q21 gain in the KRd followed by ASCT arm, whereas the PFS in patients with 1q21 gain was shorter than that in those without 1q21 gain in the KRd-only arm [31]. Thus, early ASCT could improve clinical outcomes in patients with 1q21 gain. 

Del(17p) is observed in approximately 10% of patients with NDMM and in 50–70% of MM patients with EMD and plasma cell leukemia (PCL), implying that its frequency increases with disease progression [32,33,34]. *TP53*, a tumor suppressor gene, is located at 17p13 and is monoallelic in most patients with MM. p53 haploinsufficiency induces DNA repair failure, which is related to acquired point mutations in the other *TP53* allele during disease progression. *TP53* mutations are generally observed in patients with del(17p) and MM [35]. In particular, they are detected in approximately 33% of patients with del(17p) and NDMM, and approximately 50% of patients with del(17p) and relapsed/refractory MM (RRMM) [36,37]. The frequency of *TP53* mutations is 5–8% in NDMM [36,38] and up to 25% in PCL [39,40]. In addition, mutant p53 usually has oncogenic functions, such as upregulation of the expression of *C-MYC* and genes encoding proteasome subunits [41], which can induce anticancer drug resistance [8,42,43,44]. Several new anti-myeloma agents have improved clinical outcomes in patients with del(17p) compared with old agents; however, no treatment has been reported to effectively treat MM in patients with del(17p). As a result, optimal treatment for patients with myeloma harboring del(17p) remains an unmet need [45]. Although therapeutics targeting *TP53* are also being considered in the preclinical field [46], to date, they have not been applied clinically for patients with myeloma [47]. Therefore, we believe that early therapeutic intervention before the appearance of these numerical abnormalities is important for the prevention of myeloma development with UHRCA.

Recently, a second revision of the International Staging System 2 (R2-ISS) was developed using 16 randomized clinical trial datasets by the European Myeloma Network, although clinical trials on CD38-MoAb-containing initial treatments were not included [48]. 1q21 gain/amp and del(17p) were selected as poor prognostic factors in this new prognostic model, and the frequencies of 1q21 gain/amp and del(17p) were 78% and 42%, respectively, in the high-risk group. The median PFS and OS were significantly shorter, independent of whether the initial treatment used PIs and/or IMiDs and transplant eligibility. Thus, anti-CD38 MoAb, PIs, IMiDs, and ASCT are insufficient to overcome UHRCAs considering the results of the MASTER trial and the R2-ISS.

## 3. MRD in Autografts Might Predict Clinical Outcome

In the MASTER trial, the MRD negativity rate after D-KRd induction therapy was lower in patients with UHRCAs than in the other CA groups, indicating that the myeloma cells are more frequently contaminated in the former than in the latter [6]. In two clinical studies, the MRD status of the autograft correlated with the survival time after ASCT [49,50]. Therefore, we discuss the role of MRD eradication in autografts in achieving persistent MRD negativity, particularly in patients with UHRCAs. In the MASTER trial, consecutive MRD assessments after D-KRd induction, ASCT, and four or eight cycles of D-KRd consolidation were performed using bone marrow samples. Before PIs and IMiDs were available, the presence of MRD in autografts was not associated with subsequent survival [51]. However, we considered that the contamination of myeloma cells not affecting the clinical outcomes in several previous trials because the therapeutic efficacy of conventional cytotoxic induction treatments, such as vincristine, doxorubicin, and dexamethasone, was inferior to the current induction therapy using novel agents [51,52,53,54]. 

In contrast, a single-center, retrospective analysis from Japan reported that the presence of MRD in autografts detected by next-generation sequencing (NGS) or real-time PCR predicts shorter OS and PFS after ASCT, and the lower levels of MRD in autografts are associated with longer OS and PFS [49]. In a retrospective study of patients with NDMM who underwent ASCT, those who achieved MRD negativity before ASCT had prolonged PFS compared with those who achieved MRD negativity after ASCT, suggesting that the achievement of earlier MRD negativity might be associated with a good response to induction therapy and that the incidence of MM cell contamination in the autograft might be lower in patients with MRD negativity before ASCT, leading to improved prognosis after ASCT [55]. Recently, two retrospective analyses employed next-generation flow cytometry (NGF) to reveal that MRD-negativity in autografts could predict long PFS and OS after ASCT [56,57]. Additionally, according to a retrospective analysis of MDACC, MRD-negativity was identified in patients treated with VRD induction therapy and without del17p and 1q21gain. MRD-negativity in autografts could predict long PFS and OS independent of induction therapy regimens [56]. While MRD-negativity in autografts could predict long PFS, the PFS in HRCA was short compared with non-HRCA, even in the patients with MRD-negativity in autografts [57]. Notably, the autograft MRD status could not be determined based on bone marrow samples, and the association of MRD status between autograft and peripheral blood samples was not analyzed in these two studies. Thus, the achievement of MRD negativity in autografts might be essential for longer OS and PFS, although the associated clinical significance has not yet been elucidated in large-scale prospective clinical trials in the era of novel agents.

Myeloma is generally distributed throughout the bone marrow. Therefore, residual myeloma cells may be present in untested bone marrow sites or extramedullary lesions, even when the tested bone marrow samples are MRD-negative [2,58,59,60,61]. Thus, a negative MRD status in bone marrow samples after induction therapy does not necessarily correspond to negative autografts, as MRD-positive autografts indicate the presence of circulating tumor cells (CTCs) in the peripheral blood [49,62]. Therefore, treatment strategies to eliminate MRD in the autograft include in vivo purging, which involves attacking CTCs using chemotherapy before mobilization, or ex vivo purging, which involves the positive selection of CD34-positive cells in the autograft. In several trials, ex vivo purging suppressed the contamination of myeloma cells in the autograft but did not improve survival time [53,63]. Hence, it is unlikely that a survival benefit from ex vivo purging will be achieved in the era of novel agents, considering that novel methods of ex vivo purging have not, to our knowledge, been studied on a large scale. Moreover, there is no evidence that a MoAb has been approved for in vivo purging following PI and IMiD approval. 

The incidence of myeloma cell contamination in autografts is higher in patients with HRCAs, such as del(13q), even before PIs and IMiDs were available [64]. This was demonstrated by the FORTE trial, in which the MRD-positive rate before maintenance therapy in double-hit patients was lower than that in patients with a single HRCA regardless of the treatment group [65]. Thus, patients with UHRCAs may be more likely to have myeloma cells in their autografts than patients without UHRCAs. Accordingly, we consider that it is essential to reduce the tumor burden with intensive induction therapy to obtain MRD-negative autografts, especially in patients with UHRCAs. Moreover, given that a significant association has been reported between MRD in autografts and peripheral blood, following induction therapy, MRD assessment of the peripheral blood should be performed to monitor for CTCs and prevent myeloma cell contamination in autografts [50]. Indeed, some patients with negative MRD status in autografts test positive in the bone marrow [66]. As a result, MRD assessment of autograft specimens may be the most reliable method to confirm the MRD negativity of an autograft because the MRD positivity rate in autografts might be low even in patients with positive bone marrow samples [67].

## 4. Analyzing MRD Status: Optimal Sample and Device for UHRCA

According to the International Myeloma Working Group (IMWG) criteria, the CR criteria are focused on three biomarkers: levels of monoclonal (M) protein (products from myeloma cells), distribution of myeloma, and presence of myeloma cells in the bone marrow [4]. Although MRD assessment is currently focused primarily on myeloma cells in the bone marrow, it may be more useful to consider their presence throughout the entire body, particularly after achieving MRD negativity in the bone marrow [4,61,68]. This is important because the site of bone marrow aspiration is not indicative of myeloma in the body, given that myeloma has a partial, not diffuse, distribution [69,70]. Moreover, EMD can be observed at relapse after MRD negativity [71], suggesting that myeloma cells can be independent of the microenvironment or escape harmful microenvironments even if the myeloma cell burden is reduced below the cutoff level of MRD negativity as detected by NGS or NGF. Although the technology for MRD detection has developed over time, MRD negativity cannot be considered a sign of eradication of all myeloma cells [72,73,74]. Thus, intensive treatment should be continued for myeloma cells in patients with UHRCAs as they tend to relapse aggressively owing to changes in the beneficial microenvironment for myeloma cells with 1q21 CA and the incidence of EMD in patients with 1q21 amp and del(17p) [23,26,32,33]. 

In the MASTER trial, the MRD-negativity rate as the best response was similar among SRCA, HRCA, and UHRCA; meanwhile, the PFS in the UHRCA group was shorter than those in the other groups even when MRD-SURE was achieved [74]. However, MRD assessment was performed in bone marrow samples using NGF, and patients were neither tested for myeloma disease distribution using positron emission tomography/computerized tomography (PET/CT) or magnetic resonance imaging (MRI) [69,75,76] nor for M-protein levels using mass spectrometry [77,78,79,80]. The IMWG criteria suggest combining MRD measurements using bone marrow samples with imaging MRD measurements, indicating the importance of MRD imaging to confirm the presence of extramedullary lesions [4]. Considering that myeloma cells in patients with UHRCAs frequently exhibit genomic instability and are prone to EMD complications, MRD should be analyzed using various strategies to confirm the achievement of true MRD negativity. 

## 5. Early Treatment Intervention for Myeloma

Here, the clinical features and difficulty associated with treating ultra-high-risk patients are discussed. It is important that UHRCAs are identified as late-phase myeloma because ultra-high-risk myeloma has several acquired CAs with IgH translocation or hyperdiploidy in the majority of patients [7,9,81,82]. In fact, the frequency of 1q21 gain or del(17p) is higher in symptomatic MM and RRMM than in MGUS or SMM [69]. Concerning the immunological environment in the progression of myeloma disease, NK and T cell counts reportedly increase, while their function decreases with myeloma progression [83,84]. In addition, the immunological environment may be associated with MRD status [2,85]. More specifically, the TAM, erythroblast, Treg, memory B cell, and CD4+ T cell (especially CD27+) counts in the bone marrow of MRD-positive patients are significantly higher than those in the bone marrow of MRD-negative patients [85,86]. Effector Tregs, which strongly suppress immune activity for myeloma cells, exist at the bone marrow tumor site, the abundance of which does not differ significantly from that of the peripheral blood [87,88]. Meanwhile, the exhausted T cell count is higher, and the NK cell count is lower in the peripheral blood of MRD-positive patients than in MRD-negative patients who received ASCT followed by LEN maintenance therapy [89]. In addition, the expression of KIR2DS4, which activates immunity, is lower, whereas that of NKG2A, which suppresses immunity, is higher in MRD-negative patients than in MRD-positive patients [90]. Thus, improving the immune environment can contribute to the achievement of MRD-SURE, and eradicating residual myeloma cells can balance the immune environment. As the immune environment around myeloma cells is compromised in patients with 1q21 amp, including the proliferation of TAMs and a decrease in the number of active NK cells [26], early therapeutic intervention for myelomas with a possibility of developing UHRCAs could be key to improving clinical outcomes. 

Ultra-high-risk SMM, which has an 80% or greater chance of progressing to symptomatic myeloma within 2 years, should be treated as symptomatic myeloma according to the IMWG clinical guidelines [91,92]. In addition, high-risk SMM is defined as a 50% or greater chance of progressing to symptomatic myeloma within 2 years, and has been a target of population for clinical trials [93]. Treatment strategies for high-risk SMM have two goals: preventing progression to symptomatic myeloma and achievement of durable MRD negativity [94,95]. A phase II trial of KRd induction treatment followed by LEN maintenance therapy for 2 years was conducted for high-risk SMM [95]. Preliminary data showed a complete remission rate of 70.2%, a 2-year MRD-negative rate of 77.5%, and a 90-month MRD-negative rate of 39.2%, suggesting that this treatment does not achieve durable MRD negativity in approximately two in three patients. In the GEM-SECAR trial, KRD induction and consolidation therapies and ASCT followed by lenalidomide maintenance therapy, the complete response rate was 72% and the MRD-negative rate was 68% after completion of the consolidation therapy [96,97]. Moreover, the sustained MRD negativity rate was 25.6% even 2 years after LEN maintenance therapy was discontinued [98]. After 5 years, 94% of patients had survived without relapse, suggesting that intensive treatment strategies, including ASCT, contribute to the achievement of durable MRD negativity for high-risk SMM. The 5-year PFS from this trial was indirectly superior to that from the KRd-ASCT group in the FORTE trial for NDMM, whereas the MRD-negative rates were similar in patients treated with KRD-ASCT between the trials [99]. The difference in the PFS of the KRd-ASCT groups between these trials, regardless of similar MRD-negative rates, might be attributed to a poor microenvironment and higher myeloma cell aggressiveness in MM than in high-risk SMM. Finally, the ASCENT trial (NCT03289299)—a phase II clinical trial of D-KRd induction therapy followed by DR therapy as maintenance therapy for high-risk SMM [100]—reported a 3-year PFS of 89.9% and MRD-negativity rate of 84% [101]. DARA might also be promising for high-risk SMM, although direct comparisons have not been made between ASCT and DARA for high-risk SMM. Thus, for patients with high-risk SMM, combination chemotherapy, including ASCT, may be an important strategy not only to prevent progression to symptomatic myeloma but also to achieve durable MRD negativity, leading to prolonged PFS and OS and preventing secondary high-risk CAs, such as 1q21 gain/amp and del(17p).

Additionally, cytogenetic risk may be associated with the progression of myeloma cells. The percentage of patients with 1q21 amp, del(17p), t(4;14), or t(14;16) was 59% in the KRD-ASCT group in the FORTE trial [99]. The percentage of the KRd-ASCT group with HRCA, excluding 1q21 amp, in the GEM-SACER trial was lower than that of the FORTE trial (23% vs. 34%), and the number of UHRCA patients in the GEM-SACER trial was assumed to be lower because the frequency of 1q21amp and del(17p) was higher in patients with MM than in those with high-risk SMM [96,97]. Thus, the presence of HRCAs, especially acquired CAs, could be associated with a short survival time, even if a similar intensive treatment strategy is used. Moreover, the impact of MRD positivity may differ between high-risk SMM and MM. That is, the prognosis of patients with persistent MRD-positive disease may be better than that of patients with MM who lose MRD negativity [102,103]. It is, therefore, possible that early intervention for patients with high-risk SMM without UHRCAs can achieve long-term PFS without achieving MRD negativity. However, minimal residual myeloma cells can progress to UHRCA myeloma. As a result, the achievement of durable MRD negativity is essential for UHRCA or pre-UHRCA myelomas to prevent progression or recurrence. 

## 6. Future Directions: Treatment Strategy for Patients with UHRCA Myeloma

In this section, we discuss treatment strategies for patients with UHRCAs considering the improved bone marrow microenvironment. We have demonstrated that inhibition of myeloma cell adhesion to bone marrow stromal cells, inhibition of angiogenesis, improvement of the immune environment, and improvement of bone formation are essential for the effective treatment of patients with myeloma. Therefore, a total therapeutic approach with PIs, IMiDs, anti-CD38MoAb, and ASCT is essential. The achievement of durable MRD negativity is key to improving clinical outcomes in UHRCA myeloma. Recently, the presence of CTCs, del(17p) and/or t(4;14), and T cells or NK cells with CD27 positivity were reported as predictors of MRD positivity [104]. As a result, CAs and the immune environment can be keys to developing effective treatments to achieve MRD negativity. In addition, it is essential to analyze the MRD status using not only NGF or NGS for bone marrow samples but also for peripheral blood samples, as well as imaging techniques and mass spectrometry to anticipate the possibility of EMD after treatment. Preventing myeloma cell contamination in autografts is also important, particularly in patients with UHRCAs. As a result, in vivo purging might represent effective treatment strategies, although sufficient evidence has not yet been reported for myeloma. 

Prior to the development of novel agents, two large-scale trials were conducted to assess ex vivo purging using the CD34-positive selection method; however, definitive conclusions regarding the efficacy of ex vivo purging were not obtained [53,63]. Meanwhile, DECP (DEX, etoposide, cyclophosphamide, and cisplatin) and IEV (ifosphomide, epirubicin, and etopiside) were reported as effective in vivo purging regimens [64,105]. Additionally, a phase II trial with DARA in vivo purging is ongoing [106]. Moreover, it remains unclear whether it is more beneficial to continue with the same induction therapy after a positive MRD status is achieved or alter the treatment strategy. In fact, some patients who do not achieve an MRD-negative result after four cycles of induction therapy may achieve MRD negativity by continuing with a similar therapy. For example, in the FORTE study, the MRD-negative rate increased gradually after 4, 8, and 12 cycles of KRd [99]. In the U.S. phase II trial, after 4 cycles of KRd induction therapy, followed by ASCT, 4 cycles of KRd consolidation therapy, and 10 cycles of KRd maintenance therapy, the treatment response improved during continuous CFZ+LEN therapy [102]. Moreover, in the IFM2018-04 trial, a single-arm phase II trial, patients with HRCA were treated with six courses of D-KRd, ASCT, three courses of D-KRd as consolidation, an additional round of ASCT, followed by DR maintenance therapy for 2 years [103]. The MRD-negative rate after six courses of induction D-KRd was 50%, which is potentially superior to that after four courses of D-KRd in the MASTER study [6,103]. However, although this study included peripheral blood stem cell harvest (PBSCH) with high-dose cyclophosphamide plus G-CSF with or without plerixafor after six courses of initial D-KRd, the presence of several poor mobilizers prevented sufficient PBSC for the second ASCT in the interim analysis. Therefore, PBSCH was performed after three courses of D-KRd, after which PBSC could be collected for the second ASCT, considering the retrospective evidence that more than five cycles of LEN are associated with poor mobilization. As a result, a consensus has not been reached on the optimal number of induction therapy courses; however, LEN might be administered within four months to reduce the incidence of poor mobilization. Additionally, MRD monitoring could be repeated at short intervals, and the changing of therapy might be considered for improving the MRD status in autografts, although the evidence supporting this theory is not sufficient. Furthermore, PBSCH can be carried out because the autograft MRD status can be negative in patients with MRD positivity in the bone marrow and/or peripheral blood. The incidence of myeloma cell contamination in autografts is dependent on treatment response using M-protein; the incidence of contamination is 57%, 29%, and 6% in patients with partial response (PR), very good partial response (VGPR), and CR, respectively [50]. Therefore, we posit that a VGPR or better is necessary in UHRCA patients when considering risk reduction of myeloma cell contamination in autografts. In contrast, a change or intensification of induction therapy is reasonable for UHRCA patients with a partial or worse treatment response. 

In patients with UHRCA, it may be necessary to alter the treatment strategy to eradicate minimal residual myeloma cells via a novel mechanism of action. For example, early administration of a BCMA-targeted agent to patients with MRD positivity might prevent the aggressive recurrence of the disease [107,108,109,110]. Subsequently, if MRD negativity is achieved, it would be suitable to perform PBSCH followed by ASCT. However, if MRD positivity persists despite treatment with an agent that has a novel mechanism of action, it remains unclear whether proceeding with PBSCH followed by ASCT directly or an alternative intensive re-induction therapy including cytotoxic agents, such as VTD-PACE (BOR, thalidomide, DEX, cisplatin, doxorubicin, cyclophosphamide, and etoposide), will lead to a better outcome [111].

High-dose melphalan (MEL) may be effective regardless of the autograft MRD status, as it has a completely different mechanism of action from the antitumor drugs used in induction treatment [99,112,113,114,115,116,117,118,119]. ASCT prolongs PFS in patients with MRD positivity but not in those with MRD negativity compared with VRd and KRd, suggesting that ASCT should be performed even if the MRD status is positive after intensive induction therapy; however, these results were observed in the analysis for all patients, not only those with HRCAs [99,119,120]. Meanwhile, tandem ASCT may improve clinical outcomes in patients with HRCAs [117,121]. 

Given that post-transplantation therapy has not been proven to be effective in entirely eliminating MRD [99,122], it remains controversial whether agents with novel modes of action or similar therapeutics that comprise post-transplantation therapy are more beneficial. LEN maintenance therapy until PD is a standard of care in post-transplantation settings. In the MRC-Myeloma XI and RV-MM-EMN-441 trials, LEN maintenance therapy improved the OS and PFS even after IMiDs containing induction therapy [116,123]. Additionally, in the GIMEMA MMY-3006 trial, VTD consolidation therapy improved the OS and PFS after VTD induction therapy followed by tandem ASCT [124]. However, therapeutics with alternative modes of action might be required because MRD negativity would not be achieved by repeating a similar short-term consolidation treatment with induction therapy. For instance, VRD consolidation did not improve PFS in MRD-positive patients after CVD induction therapy followed by ASCT or VMP (BOR, MEL, and prednisone) in the EMN02/HO95 trial [117]. In contrast, long-term maintenance therapy might be beneficial for conversion from MRD positivity to negativity [99,122]. In the FORTE trial, 2-year CFZ+LEN maintenance therapy increased the MRD conversion rate compared with LEN maintenance therapy alone, regardless of induction therapy of ASCT [99]. In the CASSIOPEIA2 trial, 2-year DARA maintenance therapy increased the MRD conversion rate in the VTD group but not in the D-VTD group [122]. In this second randomization of the FORTE trial, one-third of the patients who had never used LEN as induction and consolidation therapy were included. As a result, these patients did not receive LEN therapy prior to the LEN maintenance therapy. Therefore, long-term LEN-containing maintenance therapy, including LEN, which is a therapeutic with a new mode of action for the KCd (CFZ, cyclophosphamide, and DEX)/ASCT group, may increase the MRD negativity rate. Recently, among patients with UHRCA and primary plasma cell leukemia, the PFS in patients treated with twelve cycles of D-KRd +/− cyclophosphamide as consolidation followed by DR until PD as maintenance therapy in a single-arm phase II trial (the U.K. Optimum/Muknine Trial) was significantly longer than that of matched-paired patients treated with LEN maintenance therapy in the MRC Myeloma XI trial [125]. Thus, long-term post-transplantation treatment using a continuous multidrug combination or alternative agents may be essential to eradicating residual myeloma cells in MRD-positive patients.

When PIs, IMiDs, and CD38 MoAbs are used as induction therapies without achieving deep treatment responses, agents with novel modes of action might be necessary to treat patients. For example, BCMA-targeting treatments, such as chimeric antigen receptor-T cell therapy (CART) and antibody–drug conjugate (ADC) [126], could be effective options in such cases. Nevertheless, it is noteworthy that the novel BCMA-targeted CART and bispecific antibodies might be effective for patients with RRMM and triple-class refractory disease according to the subgroup analyses in a clinical trial [127]. If induction treatment cannot provide an adequate therapeutic response, conversion to BCMA-targeted agents might be an important therapeutic option to achieve MRD negativity at an early stage and in autografts. Indeed, BCMA-targeting CARTs have already been approved for RRMM [107,109], and several clinical trials on the efficacy of BCMA-targeting CARTs as a post-ASCT treatment (cilta-cel, CARTITUDE-2 trial (NCT04133636); bb2121, BMTCTN190 trial (NCT05032820), and CD19/BCMA CART (NCT03455972)) and belantamab mafodotin, a BCMA ADC (NCT04680468, NCT04876248) are ongoing (Table 1). 

The use of new molecules, such as G-protein-coupled receptor family C group 5 member D (GPRC5D), in the treatment of patients with MRD positivity after BCMA-targeted therapy may be promising. GPRC5D is a relatively specific antigen expressed on the surface of myeloma cells [128]. Both BCMA and GPRC5D are expressed in most patients with MM, although there is no significant association between *BCMA* and *GPRC5D* expression [128]. In addition, *BCMA* and *GPRC5D* expression levels are independent of disease status and refractoriness to DARA [120]. Hence, anti-GPRC5D CART is expected to be effective for patients with MM that is refractory to anti-BCMA CART as GPRC5D is expressed on myeloma cells with HRCAs, such as t(4; 14), 1q21 gain, and del(13q), excluding t(11;14) [129,130], whereas BCMA expression is low in myeloma cells with HRCAs [131]. More recently, it has been reported that superior antitumor activity is achieved using dual CART targeting both BCMA and GPRC5D compared to using a treatment that combines anti-BCMA and -GPRC5D CARTs in a mouse model [132]. GPRC5D-targeting CART and bispecific antibodies have been investigated in RRMM but not post-ASCT settings.

Fc receptor-homolog 5 (FcRH5) is a novel surface protein on MM cells. FcRH5 has a genetic locus in the chromosomal breakpoint at 1q21, the expression of which is higher in patients with 1q21 gain than in those without it [133]. Cevostamab, a bispecific antibody targeting FcRH5, has been developed and has shown promising activity against myeloma in a phase I study [134]. 

Minimal residual myeloma cells have been reported to exhibit mechanisms to escape immune activity; therefore, immune checkpoint inhibitors are considered effective treatments [86]. Pembrolizumab—an anti-PD-1 MoAb—was effective but not tolerable for RRMM in the KEYNOTE-183 trial [135]. However, several anti-PD-1 MoAb-containing regimens as a post-ASCT treatment did not show significant efficacy and exhibited toxicity; therefore, these treatment approaches were withdrawn (pembrolizumab: NCT02331368, NCT02906332, and NCT02636010; nivolumab: NCT03292263). In contrast, SLAMF7 is overexpressed in most myeloma cells [136,137]. Post-ASCT treatment, including elotuzumab (ELO)—an anti-SLAMF7 MoAb—for enhancing the immunological activity of immune cells, especially NK cells, against myeloma cells was investigated [138,139,140]. ELO has been approved in combination with LEN or POM for RRMM [141,142]. ELO treatment post-ASCT has demonstrated improved tolerability and efficacy in several clinical trials (NCT02495922, NCT03168100, NCT03003728, NCT02420860, and NCT02655458) (Table 2). 

Finally, donor immune attack might occur after allogeneic hematopoietic stem transplantation, such as up-front ASCT followed by reduced-intensity stem cell transplantation. The treatment algorithm that we have developed is presented in Figure 1.

## 7. Conclusions

The MASTER trial demonstrated that anti-CD38 MoAb, PI, and IMiDs induction therapy followed by ASCT could be effective, and treatment discontinuation could be possible in sustained MRD-negative NDMM patients without UHRCAs. However, a new treatment strategy is needed for NDMM patients with UHRCAs even if MRD negativity is achieved. For instance, MRD negativity in autograft could be associated with a long survival time, implying that purging might be one of the options for eliminating MRD in autograft. Sustained MRD negativity could be a key to long-term survival in patients with UHRCAs, and so MRD analysis using various strategies can be essential to confirm the true MRD negativity. Early treatment interventions might be essential to prevent disease progression and UHRCA acquisition. In the future, it will be critical to treat early-stage myelomas and develop therapeutics with new mechanisms of action with the goal of improving the microenvironment and preventing clonal evolution, leading to improved clinical outcomes in patients with myelomas who are candidates for UHRCA.

## Figures and Tables

**Figure 1 cancers-15-02418-f001:**
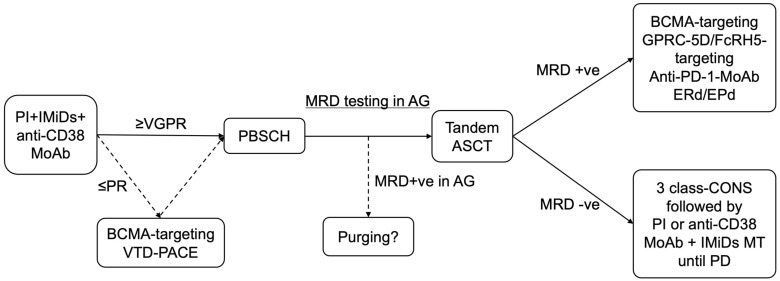
Treatment algorithm for patients with UHRCA NDMM. PI, IMiDs, and anti-CD38 MoAb-containing induction therapy are preferable options, using a proteasome inhibitor, immunomodulatory drug, and anti-CD38 MoAb, for UHRCA NDMM considering its mode of action. If VGPR is not achieved by three courses of treatment including induction therapy, alternative therapy, such as BCMA-targeting therapy or conventional chemotherapy (VTD-PACE), might be suitable to improve treatment responses, including eradication of MRD. If the MRD status in the autograft is positive, in vivo or ex vivo purging might be challenging. Intensive post-transplantation treatment is key to achieving durable MRD negativity or preventing recurrence in patients with UHRCA myeloma. If the MRD negativity is achieved, 3-class-containing combination consolidation followed by PI plus IMiDs or anti-CD38 MoAb combined lenalidomide maintenance therapy is a preferable option. If the MRD status is positive, alternative therapeutics with a new mode of action might eradicate residual myeloma cells. BCMA, GPRC-5D, and FcRH5 are new therapeutic targets. Anti-PD-1 MoAbs might effectively attack minimal residual myeloma cells as immunological escape occurs in these cells via the expression of PD-1, leading to immunological tolerance. Elotuzumab and immunomodulatory drug combination therapy might effectively enhance the immunological activity and suppress soluble SLAMF-7, the gene expression of which increases in patients with 1q21 gain/amplification. MRD, minimal residual disease; PB, peripheral blood; BM, bone marrow; PI; proteasome inhibitor; IMiDs, immunomodulatory drugs; MoAb, monoclonal antibody; BCMA, B cell mature antigen; VTD-PACE, bortezomib, thalidomide, dexamethasone, cisplatin, doxorubicin, cyclophosphamide, plus etoposide; PBSCH, peripheral blood stem cell harvest; ASCT, autologous stem cell transplantation; AG, autograft; LEN, lenalidomide; MT, maintenance therapy; CONS, consolidation therapy; GPRC-5D, G-protein-coupled receptor family C group 5 member D; FcRH5, Fc receptor-homolog 5; PD-1, programed death-1; MoAb, monoclonal antibody; ERd, elotuzumab, lenalidomide plus dexamethasone; EPd, elotuzumab, pomalidomide plus dexamethasone; PR, partial response; VGPR, very good partial response; PD, progressive disease.

**Table 1 cancers-15-02418-t001:** Clinical trials concerning BCMA-targeting therapy in post-transplantation settings.

	Drug	Phase	Study Design	Primary Endpoint
CARTITUDE-2 [NCT04133636]	Cilta-cel	2	Cilta-cel is administered after ASCT, followed by LEN (cohort D).	MRD negativity rate 1 year after cilta-cel starts
bb1212 BMTCTN190 [NCT05032820]	Ide-cel	2	Ide-cel is administered after ASCT, followed by LEN.	CR or better rate 6 months after ide-cel starts
[NCT03455972]	CD19/BCMA CART	2	Anit-CD19 (day 0) and anti-BCMA CARTs (day 1 and 2) for high-risk patients who had received ASCT.	PFS, OS, incidence of severe adverse events
[NCT04680468]	Belantamab mafodotin	2	Belantamab mafodotin is administered before and after ASCT (day 42 and 60).	MRD negativity rate 1 year after belantamab mafodotin starts
[NCT04876248]	Belantamab mafodotin	2	Belantamab mafodotin on day 1 and lenalidomide days 1–28, repeats every 8 weeks for six cycles after ASCT.	MRD negativity rate after six cycles of treatment

BCMA, B cell mature antigen; Cilta-cel, ciltacabtagene autoleucel; ide-cel, idecabtagene vicleucel; ASCT, autologous stem cell transplantation; CART, chimeric antigen receptor-T cell therapy; CR, complete response; MRD, minimal residual disease; PFS, progression-free survival; OS, overall survival.

**Table 2 cancers-15-02418-t002:** Clinical trials concerning anti-PD-1 monoclonal antibody- or elotuzumab-containing post-transplantation therapy.

	Drug	Phase	Study Design	Primary Endpoint
[NCT02331368]	Pembrolizumab	2	PEM 200 mg/kg every 3 weeks 14 days and LEN 5–15 mg/day 45–90 days after ASCT.	CR rate 180 days after ASCT
[NCT02906332]	Pembrolizumab	2	PEM 200 mg/kg every 3 weeks, LEN 25 mg/day for 14 days, and DEX 40 mg weekly for two cycles, followed by PEM 200 mg/kg every 3 weeks and LEN 15 mg/day for 14 days for two cycles.	PFS
[NCT02636010]	Pembrolizumab	2	PEM 200 mg/kg every 3 weeks for 1 year.	ORR
[NCT03292263]	Nivolumab	2	NIVO 100 mg 3 and 17 days after ASCT.	ORR
GMMG-HD6 [NCT02495922]	Elotuzumab	3	Two cycles of ELO+VRd as consolidation therapy followed by 26 cycles of ELO+LEN as maintenance therapy vs. two cycles of VRd as consolidation therapy followed by 26 cycles of LEN as maintenance therapy.	PFS
Total therapy 8 [NCT03168100]	Elotuzumab	2	ELO 10 mg/kg 1 and 15 days, LEN 15 mg 1–21 days, and DEX 20 mg weekly for 28 days, which will be alternated for 8 weeks with BOR, LEN, and DEX.	MRD status
[NCT03003728]	Elotuzumab	2	ELO 10 mg/kg on days 16, 3, 12, and 26; expanded natural killer cell infusion on day 0; and ALT-803 (interleukin-15 superagonist) 10 µg/kg on days 1, 8, 15, and 22.	Response rate
[NCT02420860]	Elotuzumab	2	ELO 10 mg/kg weekly for 2 cycles and monthly after 3 cycles of ELO and LEN on days 1–28.	PFS
[NCT02655458]	Elotuzumab	1b	ELO 20 mg/kg on day 1, LEN 10 mg on days 1–21, and autologous peripheral blood mononuclear cell (maximum number of cycles: 12).	Safety and tolerability

PD-1, programed death-1; PEM, pembrolizumab; NIVO, nivolumab; ELO, elotuzumab; LEN, lenalidomide; DEX, dexamethasone; BOR, bortezomib; ASCT, autologous stem cell transplantation; CR, complete response; PFS, progression-free survival; ORR, overall response rate; MRD, minimal residual disease.

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
