# Peer review of "Treatment Strategy for Ultra-High-Risk Multiple Myelomas with Chromosomal Aberrations Considering Minimal Residual Disease Status and Bone Marrow Microenvironment"

_cancers, 2023, doi:10.3390/cancers15092418_

Round 1

Reviewer 1 Report (Previous Reviewer 1)

this revised manuscript is more cohesive and is now appropriate for publication in this journal

well done!

Author Response

this revised manuscript is more cohesive and is now appropriate for publication in this journal well done!

-> Thank you for your response to our revised article.

Reviewer 2 Report (Previous Reviewer 3)

The paper has been improved, but still significant improvements are needed (see attached file). 

1) I find statements indicated in red incorrect, not understandable or in contrast to previous statements.

2) Yellow statements are probably mistakes. 

3) There are unnecessary repetitions, of which some even differ, probably because one was changed during revision, but not the other.  

4) The Conclusions do not reflect the text, some parts discussed are missing. I suggest to add these points to the conclusion section.

Author Response

The paper has been improved, but still significant improvements are needed (see attached file). 

-> Thank you for your response to our revised article. We revised our review article according to your recommendation as below.

1) I find statements indicated in red incorrect, not understandable or in contrast to previous statements.

-> Thank you for your pointing out our mistakes. We revised as below.

Line 194-195.

The sentence can mislead to readers, and so we deleted it.

Thus, VRD induction therapy could not improve PFS and OS after ASCT.

Line 232-234.

The sentence can mislead to readers, and so we revised as below.

Hence, MRD assessment of autograft specimens may be the most reliable means to confirm MRD negativity as the MRD positivity rate in autografts might be low even in patients with positive bone marrow samples [67].

-> As a result, MRD assessment of autograft specimens may be the most reliable method to confirm the MRD negativity of an autograft because the MRD positivity rate in autografts might be low even in patients with positive bone marrow samples [67].

Line 354-355.

The sentence can mislead to readers, and so we revised as below.

Hence, monitoring CAs and the immune environment is likely key to developing effective treatments to achieve MRD negativity.

-> As a result, CAs and the immune environment can be keys to developing effective treatments to achieve MRD negativity.

Line 360-361.

The sentence can mislead to readers, and so we revised as below.

Therefore, in vivo or ex vivo purging might represent effective treatment strategies, although sufficient evidence has not yet been reported for myeloma.

-> As a result, in vivo purging might represent effective treatment strategies, although sufficient evidence has not yet been reported for myeloma.

Line 386-388.

The sentence can mislead to readers, and so we revised as below.

Thus, a consensus has not been reached on the optimal number of induction therapy courses, however, LEN should be administered within 4 months.

-> As a result, a consensus has not been reached on the optimal number of induction therapy courses; however, LEN might be administered within four months to reduce the incidence of poor mobilization.

Line 388-390.

The sentence can mislead to readers, and so we revised as below.

Accordingly, MRD should be measured at short intervals, and switching to another therapy should be considered if the MRD status does not improve.

-> Additionally, MRD monitoring could be repeated at short intervals, and changing of therapy might be considered for improving the MRD status in autografts, although the evidence supporting this theory is not sufficient.  

Line 416-417.

The sentence can mislead to readers, and so we deleted it.

However, even if high-dose MEL can induce MRD negativity in bone marrow samples, MRD in the autograft may become increased after ASCT.

Line 418-420.

The sentence can mislead to readers, and so we revised as below.

Although it is clear that post-transplant therapy could be key to eliminating MRD from autografts [99,122], it remains controversial whether agents with novel modes of action or similar therapeutics comprising post-transplantation therapy are more beneficial.

-> Given that post-transplantation therapy has not been proven to be effective in entirely eliminating MRD [99,122], it remains controversial whether agents with novel modes of action or similar therapeutics that comprise post-transplantation therapy are more beneficial.

Line 425-428.

The sentence can mislead to readers, and so we revised as below.

However, therapeutics with alternative modes of action might again be necessary as MRD negativity will likely not be achieved by repeating a similar short-term consolidation treatment with induction therapy.

-> However, therapeutics with alternative modes of action might be required because MRD negativity would not be achieved by repeating a similar short-term consolidation treatment with induction therapy.

Line 438-440.

We mistook this sentence, and so we revised as below.

Therefore, long-term maintenance therapy, including LEN, which is a new option for the KCd (CFZ, cyclophosphamide, and DEX)/ASCT group, might increase the MRD negativity rate.

-> Therefore, long-term LEN-containing maintenance therapy, including LEN, which is a therapeutic with a new mode of action for the KCd (CFZ, cyclophosphamide, and DEX)/ASCT group, may increase the MRD negativity rate.

Line 448-450.

We mistook this sentence, and so we revised as below.

When PIs, IMiDs, and CD38 MoAbs are employed as induction therapies without achieving deep treatment responses, agents with novel modes of action might be necessary to treat patients with MRD positivity.

-> When PIs, IMiDs, and CD38 MoAbs are used as induction therapies without achieving deep treatment responses, agents with novel modes of action might be necessary to treat patients.

Line 450-453.

This sentence can mislead for readers, and so we revised as below.

For example, BCMA-targeting treatments, such as chimeric antigen receptor-T cell therapy (CART) and antibody-drug conjugate (ADC) [126], could be effective options, although there is no consensus on whether to continue or alter the induction therapy if MRD negativity is not achieved.

-> For example, BCMA-targeting treatments, such as chimeric antigen receptor-T cell therapy (CART) and antibody-drug conjugate (ADC) [126], could be effective options in such cases.

2) Yellow statements are probably mistakes.

-> Thank you for your pointing out our mistakes. We revised as below.

Line 83-86.

These two CAs are additional numerical abnormalities of IgH chromosomal translocations or hyperdiploidy, which are considered an initial event for myelomagenesis and identified less frequently in early-stage myelomas, such as monoclonal gammopathy undetermined significance (MGUS) and SMM, than in symptomatic myeloma.

-> These two CAs are occasionally detected as additional events for IgH chromosomal translocations or hyperdiploidy, which are considered an initial event for myelomagenesis and are identified less frequently in early-stage myelomas, such as monoclonal gammopathy of undetermined significance (MGUS) and SMM, than in symptomatic myeloma.

Line 98-99.

Overexpression of CKS1B necessitates an aggressive clinical course as it promotes myeloma cell growth by activating cyclin-dependent kinases.

-> Overexpression of CKS1B induces an aggressive clinical course as it promotes myeloma cell growth by activating cyclin-dependent kinases.

Line 109-111.

This finding suggests that CFZ improves the clinical outcome in patients with 1q21 gain as it represents a more potent PI than BOR.

-> This finding indicates that CFZ improves the clinical outcome in patients with 1q21 gain compared to BOR.

Line 120-124.

In a recent single-cell study, myeloma cells with 1q21 gain worsened the immune system by increasing the prevalence of tumor-associated macrophages (TAMs), decreasing the activity of effector NK cells, and increasing the gene burden of SLAMF7, which are associated with the immune microenvironment and proliferation of myeloma cells [26].

-> In a recent single-cell study, myeloma cells with 1q21 gain worsened the immune system by increasing the prevalence of tumor-associated macrophages (TAMs), decreasing the activity of effector NK cells, and the gene burden of SLAMF7, leading to an aggravating immune microenvironment and proliferation of myeloma cells [26].

Line 137-139.

Del(17p) is observed in approximately 10% of patients with NDMM, and its frequency increases with disease progression in 50%–70% of MM cases with EMD and plasma cell leukemia (PCL) [32–34].

-> Del(17p) is observed in approximately 10% of patients with NDMM and in 50%–70% of MM patients with EMD and plasma cell leukemia (PCL), implying that its frequency increases with disease progression [32–34].

Line 148-151.

Although several new anti-myeloma agents have improved clinical outcomes in patients with del(17p) compared with old agents, no treatment has been reported to effectively treat MM in patients with del(17p), remaining an unmet need for myeloma treatment [45].

-> Several new anti-myeloma agents have improved clinical outcomes in patients with del(17p) compared with old agents; however, no treatment has been reported to effectively treat MM in patients with del(17p). As a result, optimal treatment for patients with myeloma harboring del(17p) remains an unmet need [45].

Line 196-198.

The sentence can mislead to readers, and so we deleted it.

Meanwhile, according to a retrospective analysis of MSKCC, MRD-negativity was identified in patients with double-hit myeloma, not single HRCA.

Line 237-241.

According to the International Myeloma Working Group (IMWG) criteria for the diagnosis of MM, complete response (CR) is commonly used in daily practice [4]. The CR criteria are focused on three biomarkers: levels of monoclonal (M) protein (products from myeloma cells), distribution of myeloma, and presence of myeloma cells in the bone marrow.

-> According to the International Myeloma Working Group (IMWG) criteria, the CR criteria are focused on three biomarkers: levels of monoclonal (M) protein (products from myeloma cells), distribution of myeloma, and presence of myeloma cells in the bone marrow [4].

Line 280-282.

Effector Tregs, which strongly suppress immune activity in myeloma cells, exist at the bone marrow tumor site, the abundance of which does not differ significantly from that of the peripheral blood [87,88].

 -> Effector Tregs, which strongly suppress immune activity for myeloma cells, exist at the bone marrow tumor site, the abundance of which does not differ significantly from that of the peripheral blood [87,88].

Line 326.

Meanwhile, cytogenetic risk may reflect myeloma cell aggressiveness.

-> Additionally, cytogenetic risk may be associated with the progression of myeloma cells.

Line 328-330.

The property of patients with HRCA, excluding 1q21 amp, in the KRd-ASCT groups in the GEM-SACER trial was lower than that in the FORTE trial (23% vs. 34%),

-> The percentage of the KRd-ASCT group with HRCA, excluding 1q21 amp, in the GEM-SACER trial was lower than that of the FORTE trial (23% vs. 34%),

Line 338-339.

However, minimal residual myeloma cells should progress to UHRCA myeloma.

-> However, minimal residual myeloma cells can progress to UHRCA myeloma.

Line 339-342.

The sentence can mislead to readers, and so we revised as below.

Thus, the achievement of durable MRD negativity is essential for UHRCA or pre-UHRCA myelomas—high-risk IgH translocation CAs, such as t(4;14)—due to frequent acquired addition with numerous CAs, to prevent progression or recurrence.

-> As a result, the achievement of durable MRD negativity is essential for UHRCA or pre-UHRCA myelomas to prevent progression or recurrence.

Line 351-354.

We mistook the sentence, and so we revised as below.

Recently, the presence of CTCs, del(17p) and/or t(4;14), and T cells or NK cells with CD27 positivity, which are considered immune checkpoint-related antigens, were reported as predictors of MRD positivity [102].

-> Recently, the presence of CTCs, del(17p) and/or t(4;14), and T cells or NK cells with CD27 positivity were reported as predictors of MRD positivity [102].  

Line 390-392.

Meanwhile, ASCT might be performed if the autograft MRD status is negative in patients with MRD-positivity in bone marrow and/or peripheral blood.

-> Furthermore, PBSCH can be done because the autograft MRD status can be negative in patients with MRD positivity in the bone marrow and/or peripheral blood.

Line 423-425.

Meanwhile, in the GIMEMA MMY-3006 trial, VTD consolidation therapy improved the OS and PFS via VTD induction therapy followed by tandem ASCT [124].

-> Additionally, in the GIMEMA MMY-3006 trial, VTD consolidation therapy improved the OS and PFS after VTD induction therapy followed by tandem ASCT [124].

Line 435-438.

In this second randomization of the FORTE trial, one-third of the patients who had never used LEN as induction and consolidation therapy were included, suggesting that patients did not receive LEN therapy prior to the LEN maintenance therapy.

-> In this second randomization of the FORTE trial, one-third of the patients who had never used LEN as induction and consolidation therapy were included. As a result, these patients did not receive LEN therapy prior to the LEN maintenance therapy.

Line 440-445.

Moreover, among UHRCA and primary plasma cell leukemia, the PFS in patients treated with 12 cycles of D-KRd +/- cyclophosphamide as consolidation followed by DR until PD as maintenance therapy in a single arm phase 2 trial (The UK Optimum/Muknine Trial) was significantly longer than those of matched-paired patients treated with LEN maintenance therapy in the MRC Myeloma XI trial [125].

-> Recently, among patients with UHRCA and primary plasma cell leukemia, the PFS in patients treated with twelve cycles of D-KRd +/- cyclophosphamide as consolidation followed by DR until PD as maintenance therapy in a single-arm phase 2 trial (the UK Optimum/Muknine Trial) was significantly longer than that of matched-paired patients treated with LEN maintenance therapy in the MRC Myeloma XI trial [125].

Line 502-503.

Moreover, SLAMF7 is overexpressed in most myeloma cells [136,137].

-> In contrast, SLAMF7 is overexpressed in most myeloma cells [136,137].

3) There are unnecessary repetitions, of which some even differ, probably because one was changed during revision, but not the other.  

Line 382-384.

We mistook this sentence, and so we revised as below.

the presence of several poor mobilizers prevented sufficient PBSC [A1]collection for the second ASCT in the interim analysis.

-> the presence of several poor mobilizers prevented sufficient PBSC for the second ASCT in the interim analysis.  

4) The Conclusions do not reflect the text, some parts discussed are missing. I suggest to add these points to the conclusion section.

->Thank for your valuable pointing out our insufficient conclusion, and so we revised conclusion as below.

Line 548-555.

The MASTER trial demonstrated that treatment discontinuation could be possible in sustained MRD-negative patients with NDMM without UHRCAs, whereas a new treatment strategy is needed for patients with NDMM and UHRCAs, even if MRD negativity is achieved. Early treatment interventions might be essential to preventing disease progression via UHRCA acquisition. In the future, it will be important to treat early-stage myelomas with the goal of improving the microenvironment and preventing clonal evolution, leading to improved clinical outcomes in patients with myelomas who are candidates for UHRCA.

-> The MASTER trial demonstrated that anti-CD38 MoAb, PI, and IMiDs induction therapy followed by ASCT could be effective, and treatment discontinuation could be possible in sustained MRD-negative NDMM patients without UHRCAs. However, a new treatment strategy is needed for NDMM patients with UHRCAs even if MRD negativity is achieved. For instance, MRD negativity in autograft could be associated with a long survival time, implying that purging might be one of the options for eliminating MRD in autograft. Sustained MRD negativity could be a key to long survival in patients with UHRCAs, and so MRD analysis using various strategies can be essential to confirm the true MRD negativity. Early treatment interventions might be essential to prevent disease progression and UHRCA acquisition. In the future, it will be critical to treat early-stage myelomas and develop therapeutics with new mechanisms of action with the goal of improving the microenvironment and preventing clonal evolution, leading to improved clinical outcomes in patients with myelomas who are candidates for UHRCA.

This manuscript is a resubmission of an earlier submission. The following is a list of the peer review reports and author responses from that submission.

Round 1

Reviewer 1 Report

Although this paper contains a lot of very interesting information the goal/aim is somewhat obfuscated as there are multiple directions being explored.

understanding that MRD is an important technology

there paper attempts to use this as a backbone for goals of care which have not clearly been proven.  furthermore it is exploring this at multiple clinical timepoints (induction, maintenance, relapse)

The MASTER data is somewhat premature and although thought provoking, is far from mature/confirmed data

much of this manuscript seems to pull from that Trial 

also minor typo trial known as GEM-CESAR not GEM-SECAR

overall the manuscript should re-work their thoughts and either focus on the data that exists with thoughts on future directions

or switch to an entirely hypothesized paper based on minimal data in this arena

i would recommend restructuring the paper with a clear outline

Reviewer 2 Report

It is a well written report of ultra high risk patients with muliple myeloma that seem to be a group of patients that are really difficult to deal with. Until now nost of the randomized studies does not distinguish these patients, as they are not usually numerically large group of patients. But it is interesting to evaluate them and determine which combination treatment is more appropriate for this ultra high risk patients 

Reviewer 3 Report

This paper is a far too extensive disputation of the autors' ideas on treating myeloma. It does not present new data and does not focus on ultra-high risk disease (as suggested by the title). There are also some pitfalls regarding the MRD issue. For one, in the early days of autologous stem cell transplantation it has been shown that relapses tend to occur from residual tumor in the patient, not the graft and that MRD positivity of the graft is just a marker of significant residual disease in the patient. Also, MRD is assessed from blood not because it is more sensitive, but because it is easier and more comfortable for the patient than bone marrow sampling. Stating that imaging (PET and MRI) is not easily available, but NGS MRD determination is, seems odd to me, etc.